# Antiviral Approach to Cytomegalovirus Infection: An Overview of Conventional and Novel Strategies

**DOI:** 10.3390/microorganisms11102372

**Published:** 2023-09-22

**Authors:** Paolo Bottino, Lisa Pastrone, Antonio Curtoni, Alessandro Bondi, Francesca Sidoti, Elisa Zanotto, Rossana Cavallo, Paolo Solidoro, Cristina Costa

**Affiliations:** 1Microbiology and Virology Unit, A.O.U. “Città della Salute e della Scienza di Torino”, 10126 Turin, Italy; lisa.pastrone@edu.unito.it (L.P.); antonio.curtoni@unito.it (A.C.); alessandro.bondi@unito.it (A.B.); francesca.sidoti@unito.it (F.S.); elisa.zanotto@unito.it (E.Z.); rossana.cavallo@unito.it (R.C.); 2Pneumology Unit, A.O.U. “Città della Salute e della Scienza di Torino”, 10126 Turin, Italy; paolo.solidoro@unito.it

**Keywords:** HCMV, treatment, antiviral drugs, novel approach

## Abstract

Human cytomegalovirus (HCMV) is a herpesvirus capable of establishing a lifelong persistence in the host through a chronic state of infection and remains an essential global concern due to its distinct life cycle, mutations, and latency. It represents a life-threatening pathogen for immunocompromised patients, such as solid organ transplanted patients, HIV-positive individuals, and hematopoietic stem cell recipients. Multiple antiviral approaches are currently available and administered in order to prevent or manage viral infections in the early stages. However, limitations due to side effects and the onset of antidrug resistance are a hurdle to their efficacy, especially for long-term therapies. Novel antiviral molecules, together with innovative approaches (e.g., genetic editing and RNA interference) are currently in study, with promising results performed in vitro and in vivo. Since HCMV is a virus able to establish latent infection, with a consequential risk of reactivation, infection management could benefit from preventive treatment for critical patients, such as immunocompromised individuals and seronegative pregnant women. This review will provide an overview of conventional antiviral clinical approaches and their mechanisms of action. Additionally, an overview of proposed and developing new molecules is provided, including nucleic-acid-based therapies and immune-mediated approaches.

## 1. Introduction

**Biological features.** Human cytomegalovirus (HCMV), also known as human herpesvirus 5 (HHV-5), is the prototype member of the *Betaherpesvirinae* and the largest member of the virus family *Herpesviridae* [1]. It is a ubiquitous virus that infects almost all humans at some time in their lives. The virus was first isolated by three different groups of investigators: Rowe and colleagues, Weller and colleagues, and Smith simultaneously in 1956 [2]. The genome is a linear, double-stranded DNA molecule with a 236 ± 1.9 kbp size divided into a unique long (UL) and a unique short (US) region, both of which are flanked by terminal and internal repeats [3]. In detail, it contains more than 751 translated open reading frames (ORFs) of which 282 translationally active viral transcripts, 4 major long non-coding RNAs (lncRNAs) (RNA1.2, RNA2.7, RNA4.9, and RNA5.0) and at least 16 pre-miRNAs and 26 mature miRNAs [4]. Despite enclosing a much larger genome, the size of the HCMV capsid is similar to that of other herpesviruses (130 nm), structured as an icosahedral ordered nucleocapsid with triangulation number (T) = 16 and composed of 162 capsomers, divided into two distinct morphological units, 12 pentamers, and 150 hexamers [5,6,7]. Externally to the capsid, the tegument is located, which is generally thought to be unstructured and amorphous in nature. However, some structuring is observed with the binding of tegument proteins to the protein capsid. The tegument proteins are usually phosphorylated and comprise more than half of the total proteins found within infectious virions [8,9]. Finally, a lipidic bilayer envelope membrane, containing eighteen proteins including four viral G protein-coupled receptors (pUL33, pUL78, pUS27, and pUS28), covers the tegument and HCMV nucleocapsid. This envelope is similar in structure and composition to host cell membranes [10,11,12].

**Life cycle and pathogenesis.** Similar to other herpesviruses, HCMV establishes a persistent infection, remaining silent in the host and undergoing productive reactivation cycles that contribute to its efficient transmission. HCMV is able to replicate in a wide variety of cells (epithelial and mucosal cells, smooth muscle cells, fibroblasts, macrophages, dendritic cells, hepatocytes, and endothelial cells), thereby allowing for systemic spread in the human body and among host [13].

HCMV enters human cells either through direct fusion or an endocytic pathway. The virus attaches to the cell via interactions between viral anti-receptors (gH/gL/pUL128L pentamer complex, and gH/gL/gO trimer complex) and specific surface cell receptors (PDGFRα, Nrp2, and OR14I1), followed by gB activation to fuse the virus envelope with the cellular membrane [14]. Nucleocapsids are released into the cytoplasm and subsequently translocated to the nucleus, where they release viral DNA. HCMV genes are expressed in a sequential cascade, with temporal phases designated immediate-early (IE), early, and late. The major IE genes (MIE) UL123 and UL122 (IE1/IE2) are the first genes to be coded and, together with cellular host factors, coordinate the next level of gene expression (early (E) genes) involved in viral replication [15,16]. Typical early viral proteins include the DNA polymerase (pUL54), phosphotransferase (pUL97), and terminase components (pUL51, pUL52, pUL56, pUL77, pUL89, pUL93, and pUL104) [17]. Finally, HCMV encodes distinct categories of late genes, commonly referred to as leaky late (γ1) and true late (γ2): the former are expressed independently of viral DNA synthesis, while the latter are not expressed at all when viral DNA synthesis is blocked by specific inhibitors [18]. True late genes generally encode structural proteins required for the assembly of new virions, such as pUL77, pUL93 pUL115, pp28, and pp150 [19].

After DNA replication, the following steps are the encapsulation of the replicated viral DNA as capsids, which are then transported from the nucleus to the cytoplasm and coiling in the intermediate compartment of the endoplasmic reticulum (ER)-Golgi. This is then followed by a final complex two-stage envelopment and egress process that leads to virion release by exocytosis at the plasma membrane [20].

This lytic infection program leads to the release of infectious virions and can occur in an array of cells and tissues, while alternatively, in some cell types (CD14+ monocytes and their CD34+ progenitor), the virus can enter a latent life cycle that is associated with a much more limited viral transcription program and a lack of virion production [21].

**Epidemiology and transmission routes.** HCMV is a global herpesvirus highly prevalent worldwide with a prevalence of about 100% in both Africa and Asia and 45.6–95.7% in Europe and North America [22]. The heterogeneous HCMV seroprevalence appears to be related to race, ethnicity, socioeconomic status, and education level [23].

Cytomegalovirus infection can occur during pregnancy and alongside the entire lifetime along several transmission routes, as congenital HCMV infection (cHCMV) and maternal primary and non-primary infection (exogenous reinfection with a different strain or endogenous viral reactivation) of the virus during pregnancy can result in in utero transmission to the fetus (vertical road). Approximately 11% of live births born with cHCMV show abnormal clinical findings at birth: hematological disorders, cerebral malformation, chorioretinitis, and sensorineural hearing loss (SNHL), the most common sequela [24,25].

In the postnatal period, primary HCMV infections are acquired in several ways by infected fluids (e.g., saliva, breast milk, and blood products) as community exposure [26]. Breastfeeding is known to be the first close contact with a major impact, probably due to viral reactivation in the mammary glands and subsequent excretion of HCMV in milk without clinical or laboratory signs of systemic infection (negative serum IgM and negative viremia) [27]. Throughout childhood and early adulthood, HCMV is transmitted by exposure to saliva, stool, and urine [22]. Among adults, genital secretions are a common fluid for HCMV shedding, consistent with different studies that identified sexual risk factors for HCMV seropositivity or seroconversion [22,28,29]. Another important transmission route for primary infection consists of solid organ transplantation (SOT), especially in cases where there is a serological mismatch between the donor and the recipient (recipient HCMV seronegative/donor HCMV seropositive) [30]. Otherwise, infection can occur as reactivation in those patients with risk factors such as intense immunosuppression, use of lymphocyte-depleting antibodies or prednisolone, acute rejection, advanced age in the donor and/or recipient, concomitant viral infections, or genetic polymorphisms [31,32].

**Clinical features.** HCMV infection is generally asymptomatic in immunocompetent people, although clinical symptoms of primary infection may include a nonspecific glandular fever (mononucleosis) syndrome characterized by flu-like symptoms [33]. Instead, in immunocompromised or transplanted patients, HCMV primary infection or reactivation represents a fearsome complication resulting in a viral syndrome, characterized by fever and malaise as well as leukopenia, thrombocytopenia, and elevated liver enzymes. Rarely, pneumonia, hepatitis, meningoencephalitis, pancreatitis, or myocarditis could be present, requiring admission to an intensive care unit [30,34]. HCMV infection may also have indirect effects on graft dysfunction, acceleration of coronary atherosclerosis, renal artery stenosis, and the emergence of other opportunistic infections [35,36].

**Laboratory diagnosis.** Alongside serological tests and pp65 antigenemia, direct detection of HCMV DNA in clinical specimens is currently the standard method for the diagnosis of HCMV infection [2,37]. Particularly, quantitative polymerase chain reaction (qPCR) represents the gold standard, and several IVD kits based on HCMV conserved regions were developed in order to detect and quantify HCMV DNA [38,39,40,41,42]. Whole blood and plasma are the most common specimens for HCMV qPCR [43,44,45], although cerebrospinal fluid (CSF) and bronchoalveolar lavage fluid (BAL) are sometimes used [46]. Scheduled monitoring of HCMV viremia in immunocompromised individuals is pivotal to identifying patients at risk for HCMV disease, assessing preemptive therapy, and determining response to treatment [2]. Serological testing is often the reference diagnostic method for risk assessment in the prenatal/preconception field. The conventional approach of assessing the mother’s immune status using HCMV-specific antibodies entails the detection of anti-HCMV IgG and IgM antibodies as well as, in the case of IgG positivity, the measurement of IgG avidity [47]. Furthermore, the assessment of donor and recipient HCMV serologic status prior to transplantations is currently used as a marker for latent infection and the subsequent risk for donor-derived transmission and immune competence. The combination of donor and recipient serostatus allows the definition of several risk categories for HCMV disease from high-risk (serological mismatch (D+/R−)) to low-risk patients (seronegative donor and recipient (D−/R−)) [48]. In recent years, the measurement of HCMV-specific T cell activity via IFN-γ release assays (IGRAs) and enzyme-linked immunospot (ELISPOT) has gained interest for better risk stratification of immunocompromised patients or for guiding antiviral therapy. These tests can quantify HCMV cell-mediated immunity by measuring the IFN-γ that is released by CD4+ and CD8+ T lymphocytes in the presence of HCMV antigens, thus reflecting patients’ ability to control the virus and predicting the risk for post-transplant viral replication [49,50,51,52,53].

**Prevention and treatment.** The development of HCMV vaccines began in the 1970s, when the pathogenetic role of viruses on infants in utero and transplant recipients was settled, thus eliciting great interest in pharmacologic researchers. Both HCMV live vaccines (live-attenuated, chimeric, and viral-based) and non-living ones (subunit, RNA-based, virus-like particles, and plasmid-based DNA) were evaluated, but, to date, no effective candidate has been licensed. Several difficulties, such as virus immunological escape, undefined correlation with immune protection, the low number of available animal models, and insufficient general awareness, have been obstacles to the development of a satisfactory vaccine [54,55]. Moreover, epidemiological efforts should be performed in order to determine the best target populations for vaccine administration, even considering that a reduction in the birth prevalence of cHCMV and disease burden was estimated within 20 years of the start of intervention [56].

The antiviral approach for the treatment of HCMV infections relies on different drugs, such as inhibitors of viral DNA polymerase, nucleoside and nucleotide analogs, pyrophosphate analogs, and terminase inhibitors [57,58,59]. Currently, various strategies such as preemptive therapy, antiviral prophylaxis, hybrid approaches (continuous surveillance after prophylaxis for HCMV viremia with preemptive therapy), and HCMV-specific immunity-guided approaches could be used for the effective control of HCMV infection in transplanted patients [60]. However, antiviral prophylaxis and preemptive therapy are the most commonly used strategies worldwide. In antiviral prophylaxis, antiviral drugs are routinely administered to all transplant recipients at risk for HCMV disease, typically for 3 months or more immediately after transplantation, while in a preemptive therapy strategy, HCMV DNAemia is measured according to a predetermined time schedule, and antiviral drugs are administered to those transplant recipients in whom the HCMV DNA level reaches alert thresholds while the infection is still asymptomatic. Under such conditions, only a restricted cohort of patients is treated for a reduced period. [61,62]. The antiviral prophylaxis resulted in superior control of HCMV infection and prolonged time to HCMV disease in transplanted recipients without an increased risk of opportunistic infections, graft loss, drug-related adverse effects, the development of drug resistance, and mortality. For these reasons, it has become the recommended strategy by the American Society of Transplantation; however, post-prophylaxis HCMV disease (late onset of HCMV disease) remains a well-documented and widespread problem in patients receiving antiviral prophylaxis and is found to be independently associated with mortality. On the other hand, the preemptive strategy has been shown to reduce the incidence of late-onset HCMV disease and increase the HCMV-induced immune response, but it faces logistics challenges for medical centers and patient’s noncompliance with the monitoring of HCMV viremia [60,63].

This review resumes knowledge of currently available antiviral drugs and updated information on possible novel approaches to face HCMV infection.

## 2. Antiviral Approach to HCMV Infection

**Ganciclovir (C_9_H_13_N_5_O_4_).** Ganciclovir (GCV) is a synthetic nucleoside analog of 2′-deoxy-guanosin, whose primary mechanism is the inhibition of the viral DNA polymerase. The drug is primarily converted intracellularly to Ganciclovir 5′-monophosphate by a viral kinase encoded by the HCMV gene UL97 during infection. Subsequently, cellular kinases (deoxyguanosine kinase, guanylate kinase, and phosphoglycerate kinase) catalyze the formation of Ganciclovir diphosphate and Ganciclovir triphosphate. This latter is present in 10-fold greater concentrations in HCMV-infected cells than in uninfected cells [64,65]. The triphosphate form of Ganciclovir competitively inhibits the viral DNA polymerase, being inserted into the replicative HCMV genome, thus leading to premature termination and halting further viral DNA synthesis. This action disrupts the spread of HCMV and reduces the viral load.

Ganciclovir was the first antiviral agent approved for the treatment of HCMV infection, and it is mostly administered as an intravenous formulation due to its low oral bioavailability. To overcome this limit, the prodrug Valganciclovir (VGCV), a GCV valine ester, was developed. It showed higher absorption after oral administration (60.9% compared to 5.6% bioavailability of GCV), and it was rapidly metabolized to active form GCV [66]. GCV dosage in transplanted patients relies on various factors, including the type of transplanted organ, the patient’s weight, renal function, and the severity of the HCMV infection. Usually, the induction therapy dosage is between 1.25 mg/Kg and 5.0 mg/Kg every 12/24 h for 7 to 14 days, depending on patients’ age and weight, while the maintenance dosage starts from 0.625 mg/Kg every 24 h. According to the recipient’s clinical condition, the maintenance therapy phase can be performed via oral or intravenous administration. For induction and maintenance therapy, Valganciclovir dosage is between 450 and 900 mg once or twice every day, according to renal function. Meanwhile, prophylactic dosage to prevent HCMV disease relies on oral administration of 900 mg once daily [67,68]. The most common side effect associated with both Ganciclovir and Valganciclovir is leukopenia. Moreover, a recent systemic review of 102 studies (25 human and 77 animal) assessing the long-term effects in subjects receiving a prophylactic dose of Ganciclovir observed a correlation with spermatic toxic effects in those patients [69]. Despite the effectiveness of Ganciclovir and Valganciclovir, HCMV resistance can occur due to mutations targeted to the viral kinase or DNA polymerase genes. Mutations in the viral UL97 gene confer resistance to Ganciclovir, whereas mutations in the UL54 DNA polymerase gene are typically associated with high-level resistance to Ganciclovir and cross-resistance to Cidofovir and Foscarnet. Common mechanisms of HCMV resistance to Ganciclovir have been described predominantly with the UL97 mutation and occurred at codons 460–607, with mutations at codons 460 and 520 resulting in at least a five-fold increase in IC_50_ [70].

**Cidofovir (C_8_H_14_N_3_O_6_P).** Cidofovir (CDV) is a cytosine monophosphonate nucleotide analog cytosine with potent broad-spectrum antiviral activity. Unlike Aciclovir and other nucleoside analogs, which require monophosphorylation by viral kinases for activation, Cidofovir already carries a phosphonate group and does not require viral enzymes for conversion to Cidofovir diphosphate, the active antiviral compound [71]. It undergoes two stages of phosphorylation via monophosphate kinase and pyruvate kinase in order to form the active form triphosphate. Cidofovir triphosphate acts as a competitive inhibitor of HCMV DNA polymerase, encoded by the UL54 gene, preventing the incorporation of deoxycytidine triphosphate (dCTP) into growing viral DNA [72]. The active form of the drug exhibits a 25- to 50-fold greater affinity for the viral DNA polymerase, compared with the cellular DNA polymerase, thereby selectively inhibiting viral replication [73]. However, since Cidofovir is a nonobligate chain terminator, the incorporation of CDV diphosphate into the nascent strand of DNA does not necessarily result in termination [71].

Cidofovir is administered via intravenous infusion in conjunction with oral probenecid to reduce nephrotoxicity. The induction therapy relies on 5 mg/Kg once a week for two weeks, while maintenance therapy should be performed at the dosage abovementioned once every two weeks. Cidofovir was not recommended for prophylaxis. It is reserved for the treatment of Ganciclovir-resistant or refractory HCMV disease, and its use is complicated by a high rate of nephrotoxicity due to Cidofovir’s high affinity for the organic anion transporter in the convoluted proximal tubules and is responsible for cell necrosis proximal tubes [74]. UL54 DNA polymerase mutations typically add to pre-existing UL97 mutations after prolonged Ganciclovir therapy and increase the overall level of drug resistance [75].

A lipid-conjugated Cidofovir-derived prodrug, Brincidofovir (C_27_H_52_N_3_O_7_P), showed higher antiviral activity in vitro compared to Cidofovir and also in preventing HCMV reactivation. However, it failed all clinical trials due to adverse side effects, and it is not currently used [76].

**Foscarnet (CH_3_O_5_P).** Foscarnet (FOS) is a pyrophosphate analog, an oxyanion of inorganic phosphorous, that functions as a noncompetitive inhibitor of the herpesvirus DNA polymerase of all HHVs, including the most Ganciclovir-resistant HCMV isolate and Acyclovir-resistant HSV and VZV strains [73,77]. This analog acts like the pyrophosphate molecule by selectively and reversibly binding to the binding site on the HCMV DNA polymerase and inhibiting further DNA chain elongation. The role of the DNA polymerase enzyme is to cleave the pyrophosphate molecule from the DNA chain to add further nucleotides to the growing chain. Foscarnet binds and blocks that cleaving process. Although Foscarnet has selectivity for the viral DNA polymerase, it can also inhibit human DNA polymerase in much higher drug concentrations [78]. Along with Cidofovir, Foscarnet is considered a second-line agent reserved for the treatment of resistant and refractory HCMV. Foscarnet is not an orally administered drug due to low bioavailability and its inclination to be deposited within bone and cartilage. Instead, the most common route is intravenous administration, and the dosage and rate of administration are determined based on the patient’s age and weight and the specific viral infection (HCMV versus HSV or VZV). Usually, for Ganciclovir-resistant HCMV and AIDS-associated HCMV retinitis, the induction therapy is 90 mg/Kg intravenous administration every 12 h for 2 to 3 weeks or, alternately, 60 mg/Kg every 8 h for 2 to 3 weeks. Meanwhile, maintenance treatment is performed with 90 mg/Kg every 24 h or 120 mg/Kg every 24 h [64,78]. Foscarnet-associated nephrotoxicity affects 30–50% of patients after long-term use due to the deposition of drug crystals in the glomerular capillary lumen. Myelosuppression, mucosal ulcerations, and electrolyte disturbances such as hypocalcemia, hypomagnesemia, and hypophosphatemia are also common [79,80].

Foscarnet resistance mutations encountered in clinical practice are clustered in the domains of the DNA polymerase structure designated in amino terminal 2 (residues 555–600), the palm, and the finger (residues 696–981) and typically confer 3- to 5-fold decreases in antiviral susceptibility with variable low-grade cross-resistance to Ganciclovir/Valganciclovir and sometimes to Cidofovir as well [81].

**Letermovir (C_29_H_28_F_4_N_4_O_4_).** Letermovir (LMV) is a 3,4-dihydro-quinozoline that acts through inhibition of the viral terminase enzyme complex (pUL51, pUL56, and pUL89) used by HCMV in the terminal replication life cycle stage of viral DNA processing and packaging [82]. Letermovir is highly specific for HCMV, as it has no activity against other herpesviruses or any other virus, and it is one of the most potent anti-HCMV agents identified to date, with reports illustrating up to 1000 times the potency of Ganciclovir [83]. It was approved in 2017 for HCMV prophylaxis in HCMV-seropositive adult hematopoietic cell transplant (HCT) recipients and has been widely adopted in this population, but it is currently not approved for any clinical indication in solid organ transplant recipients [84,85]. The approved dosage of Letermovir is 480 mg (240 mg if co-administered with cyclosporine) once daily. Actually, it is recommended to start Letermovir at this dose in HCMV-seropositive adult patients who received an allo-HCT between days 0 and 28 and continue until day 100 post-transplantation. The route of administration is oral, and no dose adjustment is required for renal or liver impairment [86]. Letermovir is a generally well-tolerated drug, and the most commonly reported side events during clinical trials were diarrhea, nausea, and vomiting. Differently from the antiviral drugs abovementioned, it does not appear to have significant nephrotoxicity or myelosuppressive effects. Only one case of self-limiting hepatitis thought to be due to Letermovir has been reported [87].

HCMV resistance to Letermovir has emerged in both experimental and clinical settings. Mutations conferring resistance to Letermovir are most commonly mapped to UL56 (particularly codons 231–369, e.g., V236M, L241P, and R369S). Hoverer, rarer mutations of UL51 and UL89 have been implicated in the emergence of resistance. They have been described when LMV was administered as salvage therapy for drug-resistant or refractory HCMV infections and when used as a primary or secondary prophylaxis [88,89].

**Maribavir (C_15_H_19_Cl_2_N_3_O_4_).** Maribavir is a benzimidazole l-riboside antiviral compound that inhibits UL97 protein kinase and its natural substrates, thereby inhibiting HCMV DNA replication, encapsidation, and nuclear egress. This drug acts by blocking the phosphorylation of several downstream viral proteins, including UL44, thus inhibiting HCMV replication. In vitro, Maribavir is effective for HCMV and Epstein–Barr viruses but has no activity against Herpex Simplex virus 1 (HSV-1), HSV-2, Varicella Zoster virus, Human Herpesvirus 6 (HHV-6), or HHV-8 [90]. Unlike Valganciclovir/Ganciclovir, Maribavir targets a different location on pUL97 and does not require intracellular processing by pUL97 protein kinase [91].

It received U.S. Food and Drug Administration (FDA) approval in November 2021 for the treatment of adult and pediatric patients (12 years of age and older, weighing at least 35 kg) with treatment-refractory post-transplant HCMV infection/disease (with or without genotypic resistance) with Ganciclovir, Valganciclovir, Cidofovir, or Foscarnet [92]. The recommended dosage of Maribavir for both adult and pediatric patients is 400 mg orally twice daily with or without food [93]. It is 40% bioavailable, and is highly protein-bound (98%), with free plasma concentrations of Maribavir approximately 100-fold lower than total plasma drug concentrations [94].

Marabivir resistance is mediated by UL97 mutations T409M, H411Y, and C480F. They occur in patients with recurrent HCMV infection while on therapy or having no response to therapy and confer moderate (H411Y) to moderately high (T409M) Maribavir resistance, with no cross-resistance to Ganciclovir, by mapping to a hinge region of the ATP-binding site of UL97 kinase. Particularly, C480F confers the highest degree of Maribavir resistance (224-fold) of any single mutation so far encountered in vivo, along with low-grade anciclovir cross-resistance (2.3-fold) [95,96].

In Figure 1 are reported chemical structures of approved antiviral drugs for HCMV treatment.

**Tomeglovir/BAY 38-4766 (C_23_H_27_N_3_O_4_S).** Tomeglovir is a substituted 4-sulphonamide naphthalene derivative with good in vitro activity, which acts as a non-nucleoside inhibitor against laboratory and clinically adapted strains of HCMV through activity against the gene products UL89 and UL56. Its mechanism of action involves the prevention of viral DNA maturation during the replicative process by inhibition of viral DNA cleavage and capsid packaging [97,98]. It is under investigation, and preliminary studies showed antiviral effects in murine models and guinea pigs comparable to Ganciclovir [98,99]. Studies on the safety and tolerability of single oral doses (up to 2000 mg) of Tomeglovir were conducted in healthy male volunteers with no significant adverse events observed. Strains of drug-resistant HCMV generated by in vitro passage in the presence of Tomeglovir showed mutations in the UL89 and UL104 genes, suggesting that this new class of non-nucleoside compounds inhibits HCMV by preventing the cleavage of polygenic concatameric viral DNA into unit length genomes [100].

**2-bromo-5,6-dichloro-1-(β-d-ribofuranosyl)benzimidazole (BDCRB, C_7_H_3_BrCl_2_N_2_).** BDCRB and its 2-chloro homolog, 2,5,6-trichloro-1-β-d-ribofuranosyl-1H-benzimidazole (TCRB), are nucleoside analogs active against HCMV [101]. Unlike most of the currently marketed anti-HCMV agents, BDCRB and TCRB do not inhibit viral DNA synthesis, even at concentrations that completely prevent the generation of infectious virus, but exert antiviral activity by inhibition of HCMV DNA maturation. The mechanism of action is not fully understood but involves UL89 and UL56 gene products [102,103]. A study performed on guinea pig cytomegalovirus (GPHCMV) showed that the terminal structure of genomes formed in the presence of BDCRB was altered, thereby resulting in premature cleavage events and consequently in truncated genomes packed within capsids [104]. However, clinical development was not pursued after preclinical pharmacokinetic studies demonstrated that both BDCRB and TCRB are cleaved in vivo to produce the less active but more cytotoxic aglycones [102]. Within the class of benzimidazole ribosides, a derivate of BDCRB, GW275175X, exhibits similar antiviral activity without in vivo stability concerns. It acts by blocking the maturational cleavage of high-molecular-weight HCMV DNA by interaction with pUL56 and pUL89 and was advanced to Phase I clinical trial with an increasing dose of safety, pharmacokinetics, and tolerability but was later shelved to prioritize testing with Maribavir. The clinical potential of this antiviral drug still requires further study [105].

In Figure 2 are reported chemical structures of proposed terminase inhibitors for HCMV treatment.

In Table 1 are clinical details related to conventional and novel antiviral drugs.

## 3. Genome-Based Approach to HCMV Infection

**RNAi-Based Therapeutics.** RNA interference (RNAi) is an evolutionarily conserved mechanism of sequence-specific gene silencing that reduces the levels of protein products translated by a targeted mRNA [106]. The use of RNAi to reduce the levels of specific proteins not only helps to elucidate their function but also provides an opportunity to consider potential therapeutic targets that could be used to treat different diseases due to their antimicrobial activities and multiple roles in regulating gene expression. For this purpose, small interfering RNAs (siRNAs) and microRNAs (miRNAs) are the two main categories of RNAs widely investigated [107]. Since HCMV infection progresses through a well-characterized sequential process of immediate-early (IE), early (E), and late (L) viral gene expression, their product could be targeted by RNA interference approaches. To date, the available studies have investigated the role of siRNAs/miRNAs targeting the transcripts UL54 [108], UL123, and UL122 encoding the immediate-early proteins IE1 and IE2 [109,110,111], showing reduced levels of viral protein expression, DNA replication, and progeny virus production after siRNAs pretreatment.

**Ribozyme-Based Therapeutics.** Ribozymes are catalytically active RNA molecules (fewer than 100 nucleotides) or RNA–protein complexes in which solely the RNA provides catalytic activity. They are most often employed to knockdown gene expression and to inhibit infections [112]. The use of ribozymes appears to be a promising alternative to RNAi technology as no off-target hits have yet been observed [113]. They form base-pair-specific complexes and catalyze the hydrolysis of specific phosphodiester bonds, causing RNA strand cleavage. Differences exist between ribozymes in terms of size and structure, and although most naturally occurring ribozymes cleave intramolecularly at a cis linkage, the RNA component of RNase-P, which is involved in the processing of pre-tRNA molecules, acts in trans [114,115]. Several studies have been carried out to understand the catalytic mechanism and substrate binding of RNase P ribozymes and elucidated the structure of its active site, structured in a catalytic domain (C domain), with several conserved regions and a specificity domain (S domain) participating in the binding of tRNA substrate. Due to its properties, different engineered RNase-P-based ribozyme variants have been generated for the evaluation of in vitro activity towards HCMV. Kim et al. (2004) developed a functional ribozyme (M1GS RNA) that targets the overlapping mRNA region of two HCMV capsid proteins, capsid scaffolding protein (CSP) and Assemblin, which are essential for viral capsid formation. The ribozyme efficiently cleaved the target mRNA sequence in vitro, and a reduction in CSP/assembly expression levels by 85–90% was observed. Moreover, it inhibited viral growth by 4000-fold in cells that expressed the ribozyme, unlike virus-infected cells that either did not express the ribozyme or produced a ‘disabled’ ribozyme [116]. Other studies explored the efficiency of the RNase-P ribozyme variant (F-R228-IE and V718-A) towards HCMV targets such as capsid assembly protein (AP), protease, and immediate-early IE1/IE2 proteins. A 98–99% and 50,000-fold reduction was observed for protein expression and viral growth, respectively [117,118]. In all of these reports, the result suggests that the ribozyme does not interfere with host gene expression and does not exhibit cell cytotoxicity. Thus, improving the catalytic efficiency of RNase-P ribozyme could be a promising step toward developing a ribozyme-based technology for practical uses, but several limitations need to be resolved for successful delivery to targeted cells.

**Aptamer-based approach.** Aptamers are a class of nucleic acid (RNA/DNA) molecules that are beginning to be investigated for clinical use. These small molecules can form secondary and tertiary structures capable of binding cell targets [119]. Similarly to intracellular antibodies, aptamers can bind with high affinity and specificity to target proteins or mRNA under intracellular conditions and represent a powerful method to inactivate protein functions in vitro and in vivo [120]. Fomivirsen (C_204_H_263_N_63_O_114_P_20_S_20_) is a 21-nucleotide phosphorothioate oligonucleotide that inhibits HCMV replication through an antisense mechanism. Its oligonucleotide sequence (5′-GCGTTTGCTCTTCTTCTTGCG-3′) is complementary to a sequence in mRNA transcripts of the major immediate-early region 2 (IE2) of HCMV, which encodes for several proteins responsible for the viral gene expression that are essential for the production of infectious viral particles. The binding of Fomivirsen to target mRNA results in the inhibition of IE2 protein synthesis, with subsequent inhibition of viral replication [121]. It was the first in a class of antisense oligonucleotides approved by the FDA in August 1998 for the treatment of HCMV retinitis in AIDS patients who are intolerant of or have a contraindication to other HCMV regimens or who were insufficiently responsive to previous treatments for HCMV retinitis [122].

**ZFNs and TALENs.** In the 1990s, meganucleases and zinc finger nucleases (ZFNs) laid the groundwork for the concept of genome editing and initiated development in this field. These molecules comprise a chain of zinc finger proteins fused to a bacterial nuclease in order to obtain a system capable of making site-specific double-stranded DNA breaks, thus allowing gene editing. Zinc finger proteins provide site-specific targeting as they each recognize a 3–4 base pair DNA sequence [123,124]. Transcription activator-like effector nucleases (TALENs) are proteins secreted by plant pathogenic bacteria *Xanthomonas* with a core DNA binding domain of 12–28 repeats, a nuclear localization signal (NLS), an acidic domain for target gene transcription activation, and a Fok1 nuclease [125]. ZFNs and TALENs enable a broad range of genetic modifications by inducing DNA double-strand breaks that stimulate the joining of error-prone non-homologous ends or homology-directed repair at specific genomic locations [126]. One of the advantages of TALEN over ZFN is that its DNA-binding domain recognizes only one nucleotide in contrast to the three bps recognized by the first zinc finger domain of ZFN. Moreover, the TALEN system is more effective at creating double-stranded breaks and discriminating between nucleotides with different methylation states [124,127]. Targeted nucleases offer the potential to correct or disrupt gene products or sequences responsible for causing disorder, such as genetic diseases, but also to abolish the activity of viral genes [128]. This type of approach has been tested as an AIDS therapy in which ZFNs were targeted to disrupt the expression of the CCR5 gene product required by certain strains of HIV as a co-receptor to infect cells, thus resulting in HIV-1-resistant T-lymphocytes [129]. A similar approach could be applied to HCMV infection, as suggested by the discovery of the intracellular zinc finger antiviral protein (PARP13 and ZC3HAV1) and their cofactors (TRIM25) with antiviral activity. These molecules showed the ability to inhibit the replication of HCMV strain AD169 and TB40/E through the recognition of a high content of CpG dinucleotide in the viral mRNA encoding for HCMV proteins essential for virus replication. Furthermore, the production of HCMV strains from infected cells was reduced due to the destabilization of HCMV mRNA expression. However, the same results were not obtained on the HCMV Merlin strain, suggesting a specificity of the isolate for the ZFN action [130]. An interesting potential role of genetic editing could be to manage HCMV latent infection. Indeed, Chen et al. showed that three specific TALEN plasmids (MHCMV1–2, 3–4, and 5–6) were able to provide a negative regulation of latent murine HCMV infection (MHCMV) replication and gene expression through a decrease in IE1 gene expression [131].

**CRISPR/Cas9.** Clustered regularly interspaced palindromic repeats (CRISPR)/Cas9 is a gene-editing technology causing a major upheaval in biomedical research. It allows errors to be corrected in the genome and presents several promising laboratory applications such as the rapid generation of cellular and animal models, functional genomic screens, and gene therapy for the treatment of infectious diseases (e.g., HIV), malignancies, and other diseases (e.g., cystic fibrosis and Duchenne muscular dystrophy) [132]. The CRISPR/Cas-9 genome-editing mechanism contains three steps: recognition, cleavage, and repair. A designed guide RNA recognizes the target sequence in the gene of interest through a complementary base pair. While the Cas-9 nuclease makes double-stranded breaks in a site 3 base pair upstream to a protospacer adjacent motif, the double-stranded break is repaired by either non-homologous end-joining or homology-directed repair cellular mechanisms [133]. Focusing on HCMV, CRISPR/Cas-9 mechanisms also have the added advantages of inhibiting not only the reactivated virus but also the latent counterpart in cells, as well as stopping the dysregulation of innate immunity occurring during the early stages of HCMV infection. As for the RNAi and ribozyme approach abovementioned, the targeting of viral IE1 and IE2 genes led to a reduction in UL122, UL54, and UL83 transcripts with a consequential three-fold reduction in viral DNA load during both latency and reactivation phases [134]. As suggested by Natesan and Krishnan (2021), another promising target could be the major intermediate early gene promoter/enhancer MIEP/E, which is known to control all of the early genes involved in HCMV replication and reactivation from latency. Using a site-specific cleavage of the 930 bp segment, the MIEP/E gene was deleted in tested samples [135]. In another study, homologous recombination (HR) and non-homologous end-joining (NHEJ)-based methods were used to successfully mutate the HCMV genome, with optimal efficiencies of 42% and 81%, respectively, suggesting a framework for the use of CRISPR/Cas9 in a mutational analysis of the HCMV genome [136], thus also offering the high potential of genetic editing in clinical use. However, further studies are needed to fully understand the efficacy and safety of these gene-editing approaches (ZFNs, TALENs, and CRISPR/Cas-9), as well as their potential to eradicate latent infection, the major barrier for effective HCMV treatment and a long-term risk to the host.

In Figure 3 are resumed molecular mechanisms of genome-editing approaches for HCMV treatment.

**mRNA HCMV targeting**: (A) RNA interference—double-stranded RNA (dsRNA) molecules are cleaved by the DICER enzyme into short double-stranded fragments of approximately 21 to 23 nucleotides (siRNAs or miRNA). Each siRNA/miRNA is subsequently unwound into two single-stranded RNAs (ssRNA) and combined with the molecules Argonaute2 (Ago2) and heat shock protein 70/90 (HSP79/90) to form the RNA-induced silencing complex (RISC). The RISC complex then binds and degrades the target viral mRNA. (B) Ribozyme—RNA-based structure that forms base-pair-specific complexes with viral mRNA and catalyzes its hydrolysis. (C) Aptamer—RNA or DNA molecules that bind with high affinity and specificity to viral target proteins or mRNA, inhibiting their functions.

**dsDNA HCMV targeting:** (D) CRISPR/Cas9—through a guide RNA able to recognize the target sequence in the viral gene of interest, Cas9 nuclease makes double-stranded breaks, disrupting the target gene and allowing the insertion of unrelated DNA sequences. (E) TALENs—two discrete TALENs recognizing single nucleotides bind to specific sites at opposite viral DNA strands; then, an assembled FokI nuclease dimer cleaves the target viral gene. (F) ZNF—two discrete ZFNs recognizing nucleotide triplets bind to specific sites at opposite viral DNA strands; then, the FokI nuclease dimer disrupts the viral genome.

## 4. Immune-Based Approach to HCMV Infection

**Monoclonal antibodies (mAbs).** Monoclonal antibodies (mABs) are safe and effective proteins produced in the laboratory that may be used to target a single epitope of a highly conserved protein of a virus or a bacterial pathogen. They are attractive as potential therapies and prophylaxis for viral infections due to characteristics such as their high specificity and their ability to enhance immune responses [137,138]. The COVID-19 pandemic has stimulated the development of neutralizing mAbs against severe acute respiratory syndrome coronavirus 2 (SARS-CoV-2), with several mAbs authorized for emergency use. As a consequence, there has been a boost to harnessing mAbs in therapeutic and preventive settings for other infectious diseases [137]. However, only a handful of mAbs are currently approved for the management of infectious diseases (respiratory syncytial virus, *Bacillus anthrasis*, *Clostridioides difficile*, HIV-1, and Ebolavirus) [139].

MAbs anti-HCMV were developed in order to offer a safe and well-tolerated potential alternative to currently available therapies for the prevention and treatment of HCMV infection and to overcome several limitations of antiviral drugs (e.g., toxicity and development of antiviral resistance). The major antigen epitopes targeted by neutralizing mAbs are concentrated in glycoprotein B (gB) and glycoprotein H (gH). A phase 2 study evaluated the antiviral activity of CSJ148, consisting of two anti-HCMV mAbs (LJP538 and LJP539), both capable of inhibiting the function of essential viral glycoproteins. LJP538 binds to glycoprotein B (gB), while LJP539 binds to the pentameric complex (consisting of glycoproteins gH, gL, UL128, UL130, and UL131). CSJ148-treated patients showed trends toward decreased viral load, shorter median duration of preemptive therapy, and fewer courses of preemptive therapy. However, these mAbs did not prevent clinically significant HCMV reactivation in recipients of allogeneic hematopoietic cell transplants [140]. Another antibody (EV2038) was targeted to the most immunodominant epitope (AD-1) of HCMV gB. It was considered to play an essential role in gB oligomerization, which is required for gB folding and infectious virus assembly [141]. The results indicate that EV2038 is comparable or over five times more potent than other clinical candidate mAbs specific to gB and gH, also neutralizing the viral cell-to-cell spread [142]. Both studies imply that anti-HCMV mAbs can effectively limit viral dissemination, but the testing of mAbs cocktails and mAb/inhibitor combinations needs to be optimized to prevent virus dissemination and reactivation. Instead, Parsons et al. (2022) reported that a combination of anti-gH neutralizing mAbs with Ganciclovir significantly limited virus dissemination, thus supporting the use of mAbs in combination with small molecule inhibitors. Moreover, combined therapy may lower the required dose of antiviral drugs (Ganciclovir and Letermovir), reducing at the same time the associated toxicity and emergence of viral mutations [143].

**Vaccines.** The primary objectives of developing a vaccine against HCMV have been to eliminate congenital infection and to reduce morbidity and mortality in highly immunosuppressed individuals. Candidate vaccines in clinical evaluation include live attenuated, protein subunit, DNA, and viral-vectored approaches. Subunit approaches have focused on the HCMV proteins pp65 and IE-1 as important inducers of cytotoxic T cells and glycoprotein B (gB) as an important inducer of neutralizing antibodies [144,145]. DNA plasmids coding for pp65 and gB have shown preliminary evidence of efficacy in transplant recipients, with the ability to stimulate high levels of neutralizing antibodies [54]. Especially the gB surface protein combined with the MF59 oil-in-water adjuvant showed the most promising results. A regimen of three injections over a six-month period in young women naturally exposed to HCMV reduced infection acquisition, but antibodies and efficacy faded quickly. In addition, when the subunit gB protein was combined with the AS01 adjuvant that stimulates toll-like receptor 4, higher and more prolonged levels of anti-gB antibodies were elicited in humans [146]. More recently, a trial focused on the gB/MF59 vaccine observed an immune response towards the polypeptide AD-6 in more than  70% of vaccine recipients. These antibodies bind to gB and to infected cells but not the virion directly, suggesting a non-neutralizing mechanism of action and, instead, a mechanism preventing cell–cell spread of HCMV [147]. A number of HCMV vaccine candidates are currently in development, targeted to prevent congenital infection and post-transplant infections. There is evidence from Phase II trials that gB/MF59 vaccination can prevent the acquisition of HCMV in seronegative women exposed to the virus in nature, and there is solid evidence that HCMV disease in seronegative solid organ recipients and in hematogenous stem cell recipients can be prevented. However, even though no Phase III data are available yet, several candidate vaccines are moving forward. Moreover, the incorporation of epitopes derived from the pentameric complex may provide additional efficacy by inducing potent neutralizing/spread-inhibiting antibodies that target virus replication in a broad spectrum of cell types. The diversity of novel strategies being developed engenders optimism that a successful candidate will emerge [54,148].

## 5. Conclusions

HCMV infection is a significant global health problem, especially in immunocompromised individuals and congenitally infected infants. Moreover, depending on host physiology and the cell types infected, HCMV persistence comprises latent, chronic, and productive states that may occur concurrently, making its therapeutic management difficult [149]. The conventional antiviral approach relies on well-known molecules against pivotal HCMV targets. However, several side effects must be taken into consideration as well as the onset of antiviral resistance, especially in long-term therapies. For these reasons, novel antivirals against HCMV are strongly needed to treat HCMV disease. One such drug is Letermovir, an inhibitor of the viral DNA packaging, while other inhibitors (against both HCMV lytic replication and HCMV latent infection) have shown great promise [150]. Several reports suggest that RNAi, Ribozyme, CRISPR/Cas9, aptamers, and ZNFs are among the therapies that show good results, with lower toxicity, and may prove to be a promising milestone in developing therapeutic strategies for HCMV infection. However, the integration of these novel molecules in preemptive therapy, antiviral prophylaxis, or hybrid approaches must be defined in order to obtain the best clinical outcome for infected patients. Limitations to this approach include the mode of delivery, stability, and immunogenicity. Moreover, due to the ubiquity and lifelong nature of HCMV infection, efforts are addressed towards the development of any vaccine for the general population. Excellent results have been obtained, but it will likely take a very long time to eradicate the virus from the human population. The development of the vaccine could benefit from the widespread bioinformatic tools and artificial intelligence capable of predicting interaction between molecules and viral targets. Several direct anti-HCMV approaches are still available, while other drugs or molecules are currently under study with encouraging results. However, an integrated approach with immune modulation appears to be the future of controlling viral infection and managing HCMV latency/reactivation.

## Figures and Tables

**Figure 1 microorganisms-11-02372-f001:**
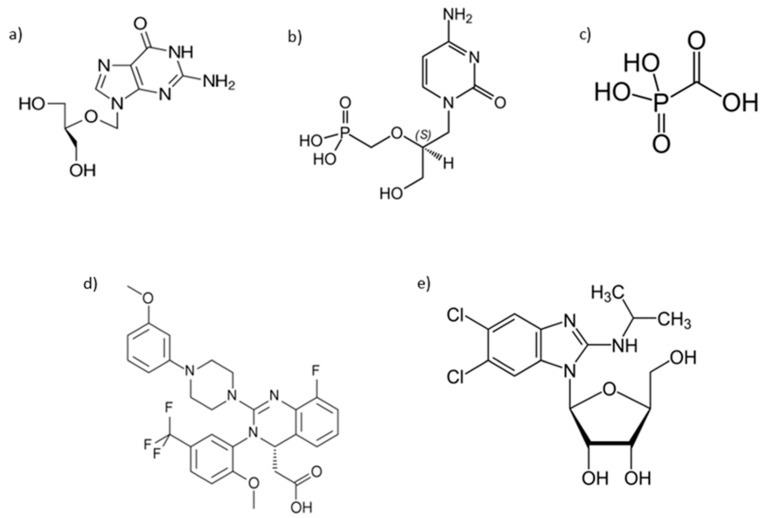
Structures of approved HCMV antivirals: (**a**) Ganciclovir; (**b**) Cidofovir; (**c**) Foscarnet; (**d**) Letermovir; (**e**) Maribavir.

**Figure 2 microorganisms-11-02372-f002:**
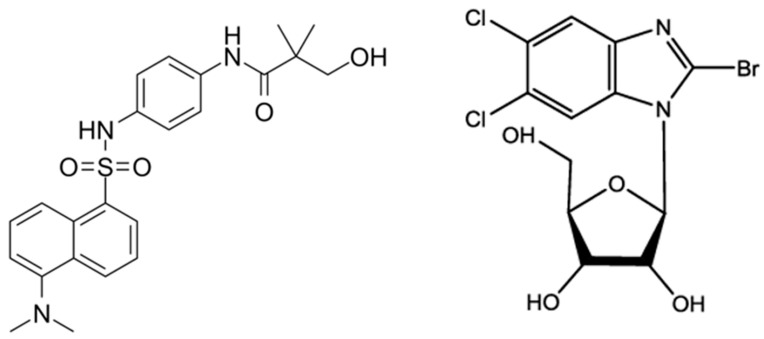
Structures of potential inhibitors of HCMV terminase complex: Tomeglovir (**left**); BDCRD (**right**).

**Figure 3 microorganisms-11-02372-f003:**
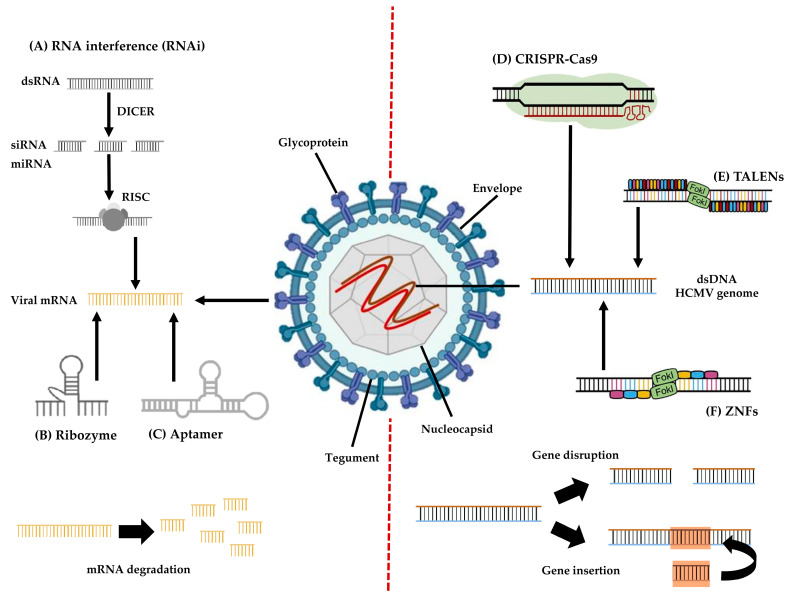
Overview of genome-editing approaches to HCMV infection.

**Table 1 microorganisms-11-02372-t001:** Clinical features of antiviral drugs for HCMV infection.

Drug	Class	Status	Commercial Name	Mechanism of Action	Route of Administration	Posology	Resistance Mechanism	Side Effects
Ganciclovir	Purine nucleoside	Clinical use (first line), FDA approval (1989)	Cytovene^®^	Competitive inhibition of viral DNA polymerase	Intravenous	Induction: 1.25 mg/Kg to 5.0 mg/Kg, twice daily (7 to 14 days) Maintenance: 0.625 to 5.0 mg/Kg die Prophylaxis: 5.0 mg/Kg die (7 days per week) of 6.0 mg/kg die (5 days per week)	Mutations on UL97 kinase and UL54 DNA polymerase genes	Bone marrow suppression (leukopenia)
Valganciclovir	Purine nucleoside, modified to improve oral bioavailability	Clinical use (first line), FDA approval (2001)	Valcyte^®^	Competitive inhibition of viral DNA polymerase	Oral	Induction: 900 mg twice daily (21 days) Maintenance: 900 mg once daily Prophylaxis: 900 mg/Kg once daily	Mutations on UL97 kinase and UL54 DNA polymerase genes	Bone marrow suppression (leukopenia)
Cidofovir	Purine nucleoside	Clinical use for treatment of HCMV ganciclovir-resistant, FDA approval (1996)	Vistide^®^	Competitive inhibition of viral DNA polymerase	Intravenous, in combination with oral probenecid	Induction: 5.0 mg/Kg, week (per 14 days) Maintenance: 5.0 mg/Kg, week Prophylaxis: not recommended	Mutations on UL54 DNA polymerase, added to pre-existing UL97kinase mutations	Nephrotoxicity
Brincidofovir	Purine nucleoside	Phase III trials, discontinued	NA	Competitive inhibition of viral DNA polymerase	Oral	NA	NA	Gastrointestinal, elevations of serum transaminases
Foscarnet	Pyrophosphate analog	Clinical use for treatment of HCMV ganciclovir-resistant, FDA approval (1991)	Foscavir^®^	Noncompetitive inhibition of viral DNA polymerase (All Herpesvirus)	Intravenous	Induction: 60 mg/Kg every 8 h or 90 mg/Kg, every 12 h (14 to 21 days) Maintenance: 90–120 mg/Kg die Prophylaxis: not recommended	Mutations on UL54 DNA polymerase, cross-resistance with Ganciclovir, Valganciclovir and Cidofovir	Nephrotoxicity, myelosuppression, mucosal ulcerations, electrolyte alterations
Letermovir	Quinazoline derivative	Clinical use, FDA approval (2017)	Prevymis^®^	Inhibition of viral terminase enzyme complex	Oral, intravenous	Induction: not Recommended Maintenance: not Recommended Prophylaxis: 480 mg once daily (0–28 to 100 days after transplantation)	Mutations on UL56, UL51 and UL89 genes	Gastrointestinal
Maribavir	Benzimidazole riboside	Clinical use for treatment of HCMV Ganciclovir, Vfoscarnetalganciclovir, Cidofovir or Foscarnet resistant, FDA approval (2021)	Livtencity^®^	Competitive inhibition of viral kinase	Oral	Post-transplant HCMV infection/disease refractory to treatment (with or without genotypic resistance): 400 mg twice daily	Mutations on UL97 kinase gene	Gastrointestinal, dysgeusia
Tomeglovir	Naphthalene derivative	Phase II trials, discontinued	NA	Inhibition of viral terminase enzyme complex	Oral	NA	NA	NA
BDCRD *	Benzimidazole riboside	Phase I trials, discontinued	NA	Inhibition of viral terminase enzyme complex	NA	NA	NA	NA

* 2-bromo-5,6-dichloro-1-(β-d-ribofuranosyl)benzimidazole; NA: not available.

## Data Availability

The data contained in this manuscript are available upon request.

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
