# Peer review of "Antiviral Approach to Cytomegalovirus Infection: An Overview of Conventional and Novel Strategies"

_microorganisms, 2023, doi:10.3390/microorganisms11102372_

Round 1
Reviewer 1 Report
To facilitate the reading, it could be useful to include a table that summarizes the different therapeutic approaches along with the different options of each of them.
Figures that summarize some of the mechanisms of action of the new therapeutic approaches would also facilitate understanding.
Specific comments:
Check the VGC dosages reported in lines 200,201 and 202.
Lines 200-201 “Valganciclovir dosage was comprised between 100 and 900 mg every day”
Line 202 “oral administration of at least 1000 mg once daily”
The commercially available presentation of VGC is in 450 mg tablets and although there is a suspension presentation, its prescription is more complex.
Author Response
We are very grateful for the advices and the comments provided by the reviewers in order to improve our manuscript. We have taken in account all these corrections in the revised version of the manuscript. Below I have stated our response to each of the suggestions made by the reviewers. We hope that the modifications made in the revised manuscript and that the way we addressed the comments made by you and the reviewers meet your expectations.
To facilitate the reading, it could be useful to include a table that summarizes the different therapeutic approaches along with the different options of each of them.
According to suggestion of both reviewers, we created a table that resumed the major clinical features of antiviral drugs, their status (clinical or preclinical), commercial name, mechanism of action, dosage / administration route, resistance mechanisms and side effects (Table 1).
Figures that summarize some of the mechanisms of action of the new therapeutic approaches would also facilitate understanding.
According to this interesting suggestion we conceptualized a figure (figure 3) that resumed all cited genome-based approaches with a brief explanation of their mechanism of action. For a better understanding we divided the figure into two sections: molecular approach towards mRNA viral target and dsDNA genome.
Check the VGC dosages reported in lines 200,201 and 202.
Lines 200-201 “Valganciclovir dosage was comprised between 100 and 900 mg every day”
Line 202 “oral administration of at least 1000 mg once daily”
The commercially available presentation of VGC is in 450 mg tablets and although there is a suspension presentation, its prescription is more complex.
We agree with the reviewer and thank for the detailed observation. We checked the reported values and modified the mistake at lines 209 – 212.
All the changes mentioned and reported in the text have been highlighted alongside the manuscript.
Reviewer 2 Report
In this manuscript, Paolo Bottino et al. provided an overview of conventional antiviral clinical approaches and their mechanisms of action. Additionally, an overview of proposed and developing new molecules is provided, including nucleic acid-based therapies and immune-mediated approaches. Overall, this is an interesting and well-written review, but I have some major and minor comments.
Major:
1. There are a substantial number of paragraphs in the introduction section of the manuscript. I suggest reducing the length of the introduction based on logical flow and coherence to ensure that readers can quickly grasp the research background and objectives.
2. It is recommended to incorporate relevant information about vaccine strategies against Cytomegalovirus infection before introducing therapeutic drugs. This will help provide a more comprehensive background.
3. It is suggested to categorize the drugs mentioned in the manuscript based on clinical and preclinical aspects and create two tables summarizing key drug information, including drug names, structures, classifications, mechanisms of action, administration routes, dosages, treatment durations, resistance profiles, and more. This will enhance the structural and readable aspects of the article.
Minor:
1. Line 304, the "FDA" is first mentioned in the manuscript, it is advisable to use the full name, "Food and Drug Administration."
2. There is a typographical error in Figure 1 where "F)" should be corrected to "d)" to eliminate confusion.
3. In certain sections, such as line 383, it is recommended to add relevant references to support your points and provide additional background information.
Minor editing of English language required.
Author Response
We are very grateful for the advices and the comments provided by the reviewers in order to improve our manuscript. We have taken in account all these corrections in the revised version of the manuscript. Below I have stated our response to each of the suggestions made by the reviewers. We hope that the modifications made in the revised manuscript and that the way we addressed the comments made by you and the reviewers meet your expectations.
There are a substantial number of paragraphs in the introduction section of the manuscript. I suggest reducing the length of the introduction based on logical flow and coherence to ensure that readers can quickly grasp the research background and objectives.
We agree with the proposed observation and thank the reviewer. In this review we provided an in-depth knowledge of biological and clinical features of HCMV infection in order to introduce several features and aspects useful to better understand some characteristics and mechanism of action of antiviral drugs. However, to facilitate the readers, we provide a partition of introduction in different paragraphs.
It is recommended to incorporate relevant information about vaccine strategies against Cytomegalovirus infection before introducing therapeutic drugs. This will help provide a more comprehensive background.
We agree with the reviewer and thank for the interesting observation. A resumed paragraph about vaccination and its difficulties was added in the background before description of antiviral approach, at lines 147 – 158.
It is suggested to categorize the drugs mentioned in the manuscript based on clinical and preclinical aspects and create two tables summarizing key drug information, including drug names, structures, classifications, mechanisms of action, administration routes, dosages, treatment durations, resistance profiles, and more. This will enhance the structural and readable aspects of the article.
According to suggestion of both reviewers, we created a table that resumed the major clinical features of antiviral drugs, their status (clinical or preclinical), commercial name, mechanism of action, dosage / administration route, resistance mechanisms and side effects (Table 1).
Line 304, the "FDA" is first mentioned in the manuscript, it is advisable to use the full name, "Food and Drug Administration."
Corrected as suggested.
There is a typographical error in Figure 1 where "F)" should be corrected to "d)" to eliminate confusion.
Corrected as suggested.
In certain sections, such as line 383, it is recommended to add relevant references to support your points and provide additional background information.
We agree with the reviewer and thank for the observation. We added a brief description of Ribozyme at lines 385-388 and a reference to line 394. Moreover, mechanisms of action of all genome-based approaches were graphical explained in figure 3 and inside its legend in order to provide a better understanding of molecular techniques potentially involved in HCMV treatment.
All the changes mentioned and reported in the text have been highlighted alongside the manuscript.